# Evidence-Based Interventions for Reducing Breast Cancer Disparities: What Works and Where the Gaps Are?

**DOI:** 10.3390/cancers14174122

**Published:** 2022-08-26

**Authors:** Rebecca D. Kehm, Adana A. M. Llanos, Jasmine A. McDonald, Parisa Tehranifar, Mary Beth Terry

**Affiliations:** 1Department of Epidemiology, Mailman School of Public Health, Columbia University, 722 W 168th St, New York, NY 10032, USA; 2Herbert Irving Comprehensive Cancer Center, Columbia University Irving Medical Center, 1130 St Nicholas Ave, New York, NY 10033, USA

**Keywords:** breast cancer, evidence-based interventions, physical activity, obesity, alcohol, tobacco control, breastfeeding, environmental chemicals, prevention and risk reduction

## Abstract

**Simple Summary:**

Breast cancer is the most common cancer in women in the United States and most countries around the world. There are many breast cancer risk factors that are amenable to intervention. This includes long-established risk factors (physical activity, obesity, alcohol consumption, breastfeeding), as well as emerging risk factors (tobacco smoke in early life, environmental chemicals). To inform future prevention, we inventoried existing evidence-based cancer control programs (EBCCP) in the National Cancer Institute’s online repository. We found that there are no existing EBCCPs for alcohol, breastfeeding, or environmental chemicals. While there are EBCCPs for physical activity, obesity, and early life tobacco control, only three programs were identified as high-quality, multilevel programs that were developed for populations that face breast cancer disparities. There thus remains a need for evidence-based interventions that can reduce breast cancer disparities.

**Abstract:**

The National Cancer Institute (NCI) has established an online repository of evidence-based cancer control programs (EBCCP) and increasingly calls for the usage of these EBCCPs to reduce the cancer burden. To inventory existing EBCCPs and identify remaining gaps, we summarized NCI’s EBCCPs relevant to reducing breast cancer risk with an eye towards interventions that address multiple levels of influence in populations facing breast cancer disparities. For each program, the NCI EBCCP repository provides the following expert panel determined summary metrics: (a) program ratings (1–5 scale, 5 best) of research integrity, intervention impact, and dissemination capability, and (b) RE-AIM framework assessment (0–100%) of program reach, effectiveness, adoption, and implementation. We quantified the number of EBCCPs that met the quality criteria of receiving a score of ≥3 for research integrity, intervention impact, and dissemination capability, and receiving a score of ≥50% for available RE-AIM reach, effectiveness, adoption, and implementation. For breast cancer risk reduction, we assessed the presence and quality of EBCCPs related to physical activity (PA), obesity, alcohol, tobacco control in early life, breastfeeding, and environmental chemical exposures. Our review revealed several major gaps in EBCCPs for reducing the breast cancer burden: (1) there are no EBCCPs for key breast cancer risk factors including alcohol, breastfeeding, and environmental chemical exposures; (2) among the EBCPPs that exist for PA, obesity, and tobacco control in early life, only a small fraction (24%, 17% and 31%, respectively) met all the quality criteria (≥3 EBCCP scores and ≥50% RE-AIM scores) and; (3) of those that met the quality criteria, only two PA interventions, one obesity, and no tobacco control interventions addressed multiple levels of influence and were developed in populations facing breast cancer disparities. Thus, developing, evaluating, and disseminating interventions to address important risk factors and reduce breast cancer disparities are needed.

## 1. Introduction

The National Cancer Institute (NCI) has established a repository of evidence-based cancer control programs (EBCCP) and has increasingly called for the usage of these EBCCPs to reduce the cancer burden. To be included in the repository, each EBCCP must have been published in a peer-reviewed journal; produced one or more positive behavioral and/or psychosocial outcome in a research study using an experimental or quasi-experimental design; and, unless otherwise copyrighted by the program developer, make available the program materials that include English for the larger cancer control community. Breast cancer is the most common cancer in women in the United States (U.S.) and the world, with increasing incidence in many populations including among U.S. women <55 years of age [1]. There are persistent breast cancer disparities across racial, ethnic, socioeconomic, and geographic lines. Some noted examples in the U.S. include a higher incidence of late-stage cancers in non-Hispanic Black (NHB) and Hispanic women relative to non-Hispanic White (NHW) women and incidence of one of the most aggressive subtypes of breast cancer (triple-negative breast cancer [TNBC]) is double in NHB compared with NHW women [2,3,4].

Some of the established risk factors for breast cancer may be amenable to modification and public health guidelines have emphasized reduction of the cancer burden through lifestyle modification. For example, the American Cancer Society (ACS) lifestyle prevention guidelines recommend ≥150 min of moderate intensity physical activity (PA) per week, consuming ≤1 alcoholic drink per day, and maintaining a body mass index (BMI) of <25 kg/m^2^. In addition to strong evidence supporting these recommendations in the general population, epidemiologic data suggests that adherence to these three recommendations is associated with 44–53% lower overall mortality in women with a family history and those with a personal history of breast cancer [5]. While these recommendations have been in place since the early 1980s [6], data show that most women are unaware of these guidelines, particularly with respect to alcohol consumption [7]. In addition to these lifestyle factors, the American Institute for Cancer Research (AICR) Continuous Update Project (CUP) concluded that there is strong evidence for a probable association between breastfeeding and decreased breast cancer risk in the mother [8,9]. There is also growing evidence that active and passive tobacco smoke exposure in early life increases breast cancer risk, particularly in premenopausal women [10], and data are accumulating linking other environmental chemical exposures to increased breast cancer risk [11,12,13]. Thus, evidence-based interventions for emerging risk factors may also reduce the breast cancer burden. 

Focusing on intervention programs (hereafter programs) specific to reducing the risk of breast cancer, we inventoried and summarized NCI EBCCPs and identified where possible gaps remain. We highlight programs that are of high quality and with implementation potential, particularly those that were implemented across multiple levels of influence (i.e., individual, interpersonal, community, and society) using the National Institute of Minority Health and Health Disparities (NIMHD) research framework and were developed in populations representative of those that face breast cancer disparities (e.g., racial and ethnic minorities, medically underserved groups [14])—groups for whom EBCCPs must be tailored and effective in order to address persistent disparities. We close with a discussion of gaps in the existing programs and highlight additional modifiable factors that may be amenable to novel interventions that aim to reduce the unequal breast cancer burden.

## 2. Materials and Methods

### 2.1. Selection of Risk Factors

We downloaded available data for interventions relevant to reducing breast cancer risk (based on the criteria described below) directly from the NCI query tool for EBCCP (https://ebccp.cancercontrol.cancer.gov/, (accessed on 27 April 2022)). We restricted our analysis to programs related to established risk factors (PA, obesity, alcohol, breast feeding) or risk factors with growing evidence (tobacco smoke in early life, environmental chemicals). We excluded programs focused on early detection (screening), psychosocial interventions in cancer patients/survivors, and interventions only recommended to high-risk individuals (informed decision making about genetic testing, risk reducing surgeries and chemoprevention). 

### 2.2. Quantitative Analysis of EBCCPs

The downloaded data included program titles and descriptions, program area (e.g., PA, obesity), population focus (e.g., age, sex, race, ethnicity), delivery location (e.g., home, school, health care setting, religious establishments), community type (e.g., suburban, urban), and availability of intervention materials, purpose of the program, and program URL. In addition to the directly downloadable data, the NCI EBCCP website includes summary metrics scored by external peer reviewers who are topic experts as described on the website. These include the four metrics of the Reach, Effectiveness, Adoption, Implementation, and Maintenance (RE-AIM) framework; programs were assigned a percentage from 0–100% for each metric (where applicable). The ratings for research integrity, intervention impact, and dissemination capability were also included, which were each assigned a score from 1 to 5 (1 = low to 5 = high) [15]. The scoring summary for these constructs is in Appendix A. Using this information, we quantified high-quality EBCCPs as those meeting the following criteria:(1)Independent score of ≥3 for research integrity, intervention impact, and dissemination capability.(2)Independent score of ≥50% for reach, effectiveness, adoption, and implementation.

We assessed how many of the high-quality EBCCPs were multilevel (i.e., designed to address two or more levels of influence including the individual, household/family, school, clinic, and community levels) and were originally developed and implemented in populations facing breast cancer disparities. This includes diversity across race and ethnicity (defined as study population ≤50% NHW participants), socioeconomically disadvantaged participants, and/or those from medically underserved or rural geographic areas. We determined if the EBCCP was multilevel and developed in a population facing breast cancer disparities by reviewing the primary publications and, in some cases, the secondary publications listed on the NCI EBCCP website.

## 3. Results

Table 1 summarizes the overall categories of risk factors considered and the available number of NCI EBCCPs.

Physical activity: There were 41 NCI EBCCPs addressing PA. Of these, 20 (49%) focused on children or adolescent populations. This included 17 school-based or after-school/organization-based (YMCA, Girl Scouts) programs, one clinic-based program, and two home-based programs. Fourteen EBCCPs focused on adults between the ages of 18 and 64 years, including six faith-based or community-based programs, two clinic-based programs, two employee-based programs, and four individual-based programs for specific subgroups (female college students, sedentary adults, obese adults, active smokers). Four EBCCPs focused on older adults, defined as 65 years and older, which were designed as either home-based or facility-based programs. Three EPCCPs were community-level interventions, focused on park improvements or media campaigns to promote PA. Twenty-four percent (*n* = 10) of PA EBCCPs met all the quality criteria. Of these, 6 were school-based programs for youth, one was a web-based program for female college students, two were employee-based programs for adults, and one was a group-based program for older adults with osteoarthritis. Three of the 10 high-quality EBCCPs were multilevel interventions (targeted physical activity at the school, family, and individual levels), all of which were youth programs and two [16,17] of which were developed and evaluated in populations facing breast cancer disparities (Table 2). 

Obesity: There were 29 NCI EBCCPs addressing obesity. More than half (*n* = 16) targeted children and/or adolescents, were implemented in school-based or clinic-based settings, and focused on decreasing anthropometric and body composition measurements (e.g., weight, BMI, percent body fat), to maintain a healthy weight, or to reduce obesity-related comorbidities (e.g., hypertension) through dietary modifications and/or increasing PA engagement. The remainder (*n* = 13) focused on young adults (ages 19–39 years), adults (40–65 years), and/or older adults (65+ years) within work-based or other settings (e.g., religious institutions). Of these programs, one focused only on young adults, one focused only on adults, three focused only on older adults, three focused on young adults and adults, and five focused on young adults, adults, and older adults. One EBCCP specifically targeted overweight and obese breast cancer survivors [18] and one program targeted adults at increased risk for colorectal cancer [19]. Besides the one program targeting breast cancer survivors [18], none focused on populations at increased risk for breast cancer or, to the best of our knowledge, included specific messaging about the impact of obesity on breast cancer risk. Five (17%) obesity EBCCPs met the quality criteria. Among them, only one program [20] (3%) considered more than one level of influence (individual, parent/family, and school, and school meal supplier levels) and was developed and evaluated in a population facing breast cancer disparities.

Alcohol: No NCI EBCCPs currently exist for alcohol intake. Despite there being several dietary EBCCPs, according to the AICR CUP [8], there are no conclusive epidemiologic data relating specific dietary factors (except alcohol) to breast cancer risk. 

Breastfeeding: No currently existing NCI EBCCPs focus on the role of breastfeeding and breast cancer risk reduction. 

Tobacco control in early life: There were 28 NCI EBCCPs addressing tobacco control, of which 16 were either designed and delivered to adolescents only (*n* = 11) or included adolescents (*n* = 5; 11–18 years), with one program additionally including children (<11 years). The programs ranged in focus from promoting smoking prevention and cessation behaviors to reducing tobacco use in the household or communities, and a majority were conducted in one or more of the following: school (*n* = 8), home (*n* = 6), or clinical (*n* = 4) settings. Thirteen of the 16 programs (81%) were identified as high-quality, two of which were directed at multiple levels of influence. However, neither of these high-quality, multilevel programs were developed in populations facing breast cancer disparities. 

Environmental chemical exposures: No NCI EBCCPs currently exist for reducing risk from environmental chemical exposures.

**Table 1 cancers-14-04122-t001:** Summary of breast cancer risk reduction categories considered.

Risk Factor	Evidence from National Cancer Institute (NCI) [21] Recommendations for Breast Cancer Prevention	Evidence from American Institute for Cancer Research Continuous Update Project Findings [8]	Number (#) of High-Quality ^a^, Multilevel EBCCPs in a Population Facing Breast Cancer Disparities ^b^/# of EBCCPs Meeting the Quality Criteria/# of NCI EBCCPs
Physical Activity	Decreases risk	Strong evidence of decreased pre and postmenopausal risk.	2/10/41
Higher Body Fatness in Young Adulthood	Not discussed	Probable evidence of decreased pre and postmenopausal risk.	1/4/16
Adult Body Fatness (marked by BMI, waist circumference, and waist-hip ratio) and Weight Gain in Adulthood	Increases risk	Strong evidence of increased postmenopausal breast cancer risk.	0/1/13
Alcohol	Increases risk	Strong evidence that alcohol increases pre and postmenopausal breast cancer, no strong evidence for other dietary factors.	0
Tobacco Exposure in Early Life	Not discussed	Not discussed in report.	0/13/16
Breastfeeding	Reduces risk	Probable evidence of decreased pre and postmenopausal risk.	0
Environmental Chemical Exposures	Not clear	Not discussed.	0

^a^ High-quality was defined as receiving an independent score of ≥3 for research integrity, intervention impact, and dissemination capability; and receiving an independent score of ≥50% for reach, effectiveness, adoption, and implementation. **^b^** Health disparities population defined as a multiracial/multiethnic population with ≤50% non-Hispanic White, a socioeconomically disadvantaged group, and/or a medically underserved or rural geographic area.

**Table 2 cancers-14-04122-t002:** Summary of high-quality, multilevel NCI EBCCPs developed and evaluated in populations facing breast cancer disparities.

Risk Factor	Program Title	Age Group	Delivery Location	Program Description	Research Integrity	Intervention Impact	Dissemination Capability	Reach	Effectiveness	Adoption	Implementation
Physical activity	New Moves [16]	11–18 years (adolescents)	Schools	All-girls physical education classes combined with individual coaching sessions and personal goal setting.	4.3	3.0	5.0	100%	66.7%	100%	62.5%
Physical activity	Alberta Project Promoting active Living and healthy Eating (APPLE Schools) [17]	0–10 years (children)	Schools	Full-time school health facilitator implemented healthy eating and active living strategies while addressing the unique needs and barriers to health promotion in the school environment by engaging all stakeholders, including parents, staff, and the community.	4.2	3.0	4.0	80.0%	66.7%	66.7%	57.1%
Obesity	5-a-Day Power Plus [20]	0–10 years (children)	Schools	School-based, multi-component intervention aimed at increasing fruit and vegetable consumption among fourth- and fifth-grade students through four intervention components: (1) behavioral curricula for fourth and fifth grade students, (2) parental involvement/education, (3) school food service changes, and (4) industry involvement and support.	4.1	3.9	4.5	80%	66.7%	100%	71.4%

## 4. Discussion

Our review revealed major gaps in NCI EBCCPs that may be relevant for reducing the breast cancer burden: (1) there are no evidence-based programs for key breast cancer risk factors including alcohol, breastfeeding, and environmental chemical exposures; (2) of the interventions that do exist for PA, obesity, and adolescent tobacco use, only a small fraction (24%, 17% and 31%, respectively) were deemed high-quality, with limited high quality programs being dual multilevel interventions and developed in populations with greater breast cancer disparities (two PA and one obesity programs); and (3) there is a paucity of interventions across the breast cancer control continuum. Caveats to our review include that we restricted our analysis to interventions included as part of the NCI’s EBCCP program and did not include evidence-based interventions from elsewhere. While we note that other evidence-based interventions relevant to cancer risk factors may be described by other institutes and organizations such as National Institute of Alcohol Abuse and Alcoholism (NIAAA), National Heart Lung and Blood Institute (NHLBI) and the Center for Disease Control and Prevention (CDC), these are not subject to the same evaluation process and available in a searchable online database like NCI’s EBCCPs. We also primarily focused on the initial publication for each program to evaluate whether they were originally developed and evaluated in populations facing unequal breast cancer burden given that impacting disparities require programs that specifically address the complex and unique lived experiences, as well as the many contextual and structural barriers to achieving health, in these groups. However, we acknowledge that some of the programs may have subsequently been adapted and/or tested in other populations.

### 4.1. Absence of EBCCPs for Key Risk Factors

There is a need for EBCCPs for reducing alcohol exposure given the increasing prevalence of alcohol consumption, its established association with breast cancer risk, and the recommendations from the American Public Health Association and American Society of Clinical Oncology supporting policies and strategies to reduce alcohol for breast cancer risk reduction [7,22,23]. Additionally, emerging data suggest that binge drinking (beyond just regular alcohol consumption) may independently increase the risk of breast cancer [24] and binge drinking rates are also on the rise [25]. Further, there are no EBCCPs for increasing breastfeeding even though we know that breastfeeding rates are different across racial and ethnic subgroups [26,27]. 

Although there are some NCI EBCCPs related to early life tobacco control, there are none that address interventions during key windows of susceptibility [12]. Further, no existing EBCCP focuses on reducing environmental chemical exposures to carcinogens or other toxic chemicals such as endocrine disrupting chemicals, which may have significant long-term health effects particularly for breast cancer [12]. For example, prenatal exposure to some chemicals found in personal care products, including hair products, is associated with earlier menarche [28,29]—a key risk factor for breast cancer. Furthermore, emerging evidence supports an association between hair dye and relaxer use and breast cancer risk and breast cancer clinicopathology [30,31,32], particularly among NHB women who use more hair products per capita than any other racial and ethnic group. Environmental exposures are a plausible driver of breast cancer disparities especially given that the cancer burden is higher in the same neighborhoods that have higher exposure to environmental contaminants [33,34,35], and in households and individuals using products high in chemical exposures [36,37,38] and with heightened vulnerability to deleterious health effects of these environmental toxicants [39,40]. Thus, there is a need for including [41] and developing additional effective and scalable interventions to reduce the breast cancer burden from environmental chemical exposures. There is also a need for EBCCPs on genomic susceptibility that can identify women who may be at increased risk of the detrimental health effects of these environmental chemical exposures. NCI currently includes only two genomic susceptibility EBCCPs (one focused on breast cancer susceptibility and the other on colorectal cancer susceptibility), neither of which met our quality metrics, nor were they inclusive or multilevel. This is of concern for breast cancer disparities given that NHB women develop breast cancer earlier than NHW women, despite having similar prevalence of pathogenic mutations in key susceptibility genes and genomic susceptibility of one family member impacts the entire family [42]. 

### 4.2. Lack of Multilevel Interventions and Other Gaps

This review revealed that while there are several NCI EBCCPs addressing PA, an established risk factor for breast cancer, only two were high-quality, multilevel programs developed in populations facing breast cancer disparities [16,17]. This is concerning given that racial and ethnic minority populations have lower levels of PA across the lifecourse than NHW populations and experience a greater decrease in PA between adolescence and early adulthood [43]. Multilevel PA EBCCPs are essential for reducing breast cancer disparities given that populations facing breast cancer disparities consistently confront barriers to PA at multiple levels of exposure, including the individual (e.g., lack of resources and knowledge), interpersonal (e.g., competing family responsibilities, lack of social support), and community levels (e.g., lack of access to safe parks and affordable recreational facilities) [44]. We also note the lack of PA EBCCPs developed specifically for populations at increased breast cancer risk, despite evidence supporting that PA reduces risk and improves outcomes after breast cancer diagnosis in those with increased familial or genetic risk of disease [45,46]. There are also few PA EBCCPs that have been specifically developed for women in early adulthood, the period in life when PA levels often dramatically decrease [43], which might be particularly important for addressing the rise in breast cancer incidence that has occurred in young women over time [1]. We identified only one high-quality PA EBCCP that was designed specifically for young adult women—specifically college/university students, a population typically of higher socioeconomic position—and evaluated in a predominantly NHW population [47]. There thus remains a need for high-quality multilevel PA EBCCPs in populations facing breast cancer disparities.

Having a BMI ≥30 kg/m^2^ in adulthood is another established risk factor for several cancers, including postmenopausal ER+ breast cancer. Current evidence suggests that obesity is also associated with a greater risk of TNBC and premenopausal ER- breast cancer [48,49,50,51,52,53,54,55,56,57,58,59,60,61] and a lower risk of premenopausal ER+ disease [49,50,53,58,61,62,63]; however, some studies have reported no significant association between obesity and premenopausal ER+ breast cancer overall [48,57] or specifically among NHB women. Emerging evidence supports a relationship between obesity and poorer response to neoadjuvant chemotherapy [64,65,66] and with poorer breast cancer outcomes irrespective of menopausal status [67,68]. Yet, elevated BMI (which is widely used as a proxy for total adiposity) is inconsistently associated with breast cancer mortality across racial and ethnic groups [69,70], while elevated waist-to-hip ratio (WHR) is more consistently associated with poorer outcomes. This suggests that perhaps evidence-based interventions focusing only on reducing BMI as a means for cancer risk reduction may be insufficient among diverse populations due to differences in body composition at a given BMI across race and ethnicity, which contributes to the misclassification of adiposity. Given the disproportionate burden of both higher levels of adiposity and poorer breast cancer outcomes among some racial and ethnic minority groups, relative to their NHW counterparts, interventions targeting adiposity and body composition measures (e.g., WHR, percent body fat, fat mass index) rather than BMI alone, as well as those targeting the metabolic and inflammatory imbalances associated with adiposity-related cardiometabolic comorbidities might be an important aspect of reducing persistent breast cancer disparities that needs greater attention. As shown, only five obesity EBCCPs [16,20,71,72,73] met the quality metrics and four of them targeted childhood and adolescence [16,20,71,72]. There was one high-quality program for youth that was multilevel and included a sample with >50% of individuals from populations facing breast cancer disparities [20], but there were none for adults. The lack of high-quality obesity programs in adult populations is a major gap given that postmenopausal women might benefit most in terms of breast cancer risk reduction from such programs, and multilevel interventions might prove more effective for groups with a greater prevalence of excess adiposity [74,75,76]. Youth programs targeting obesity are also important given that obesity tracks from childhood to adulthood [77,78]. However, understanding the full impact of early life obesity programs on breast cancer risk reduction is complicated by the fact that body fatness in early adulthood has been associated with decreased breast cancer risk [8]. 

In addition to a lack of high-quality, multilevel EBCCPs for key risk factors that are important for breast cancer, there is a paucity of EBCCPs that address other relevant areas in breast cancer risk reduction. For example, although some established risk factors, such as PA and maintaining a healthy body weight, are shown to improve outcomes after diagnosis [46,79,80,81], including reduced risk of breast cancer recurrence and new primaries [81,82], most programs were designed to reduce the risk of the first cancer and do not target cancer survivors. Instead, interventions for cancer survivors largely focus on psychosocial interventions, which are important but might be complemented by lifestyle interventions. 

## 5. Conclusions

We recommend the development of valid and scalable, multilevel interventions developed and evaluated in populations facing breast cancer disparities that address established risk factors to reduce breast cancer disparities. Given that the etiology of breast cancer is complex and multifactorial, future programs are needed that move beyond the conventional approach of targeting individual risk factors in isolation and instead focus on multiple risk factors simultaneously. We further recommend the prioritization of resources for etiologic research on breast cancers that are more aggressive and have fewer established modifiable risk factors and treatment options (e.g., TNBC), which will inform future development of interventions. For example, a greater understanding of the mechanisms through which adiposity impacts breast cancer risk and prognosis might inform future clinical and translational studies that integrate alternative measures of adiposity (as opposed to BMI alone) to enhance risk reduction strategies for greater effectiveness across racial and ethnic groups and tumor phenotypes. The gut microbiome might be another important area of research, given that it is potentially modifiable, might play a role in weight gain and weight loss, and has been shown to differ between obese and non-obese individuals [83,84]. Thus, development of novel interventions that incorporate adiposity and the microbiome should be considered for cancer risk reduction. Finally, for both etiologic and intervention research, there is a need for research beyond individual lifestyle factors including a focus on community and societal level environmental and social factors as individual lifestyle risk factors only explain a portion of the cancer burden.

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
