# Peer review of "Evidence-Based Interventions for Reducing Breast Cancer Disparities: What Works and Where the Gaps Are?"

_cancers, 2022, doi:10.3390/cancers14174122_

Round 1

Reviewer 1 Report

This paper reports on a review of the National Cancer Institute's evidence-based cancer control programs (EBCCP) for breast cancer and provides a systematic approach to the evaluation of existing research captured by the EBCCP and gaps for identified risk factors.  The intent is to focus on the quality of existing research using defined criteria and on directing attention to the gaps.

General Comments

1. The paper is well written and provides a useful review for breast cancer investigators interested in understanding where further research is needed to improve the quality of evidence or  to fill gaps where they exist.

2. Some attention to definitions and terminology may be helpful: a. 'Program' The authors are constrained by the terminology used by then NCI in naming the EBCCP, but are these really programs as might be expected in a program project, SPORE or center, rather than traditional research projects? Clarification of the meaning of 'program' in the EBCCPs is needed, b. 'Multilevel'. This term is used throughout the paper to describe interventions at the 'individual, interpersonal, community, and society' but it is not clear how the authors actually assessed program or project activities across all these levels. Clarification would help c. 'Health Disparities Populations'. I realize that this is the term used in the legislation (ref 14) and serves as a sort of shorthand,  but it is an awkward ungrammatical term that is meant to describe populations in which differences in health (cancer) outcomes or determinants have been observed and where health inequities are of concern. The author's term 'populations that face breast cancer disparities' or inequities is more satisfying as long as it is not used repeatedly.

3. I question the strategic wisdom of suggesting that every risk factor needs an EBCCP (e.g., an EBCCP on breast feeding). This is a reductionist  and 'siloed' approach to a disease that is actually rather complex etiologically with multiple interactions over the lifecourse. Can the authors justify their or the NCI's motivations for such an approach?

Specific Comments:

4.There is repetition between the last paragraph of the Introduction and paragraph 2.1 in the Methods. Would be best to put mehtods in the Intro into Methods.

5. In the footnote to Table 1 it refers to 'Health disparities population' as defined by greater than or equal to 50% non-Hispanic White. Shouldn't that be less than 50% White? Check that.

6. Some of the language is overly dramatic. section 4.1 'grave concern', 'critical' this and that.

7. page 8, sect 4.2. The authors mention that there is no or little focus on cancer survival by EBCCPs, but everything preceding in Methods refers only to etiologic risk factors. If this is the intended focus of the EBCCPs it seems inappropriate to call it out as a gap.

8. Not a criticism or need for comment, but the discussion of the challenges faced by the assessment of body habitus and the weakness of the BMI as well as mention of the evolving challenges of understanding the influence of the microbiome and the need to move for research beyond the individual level to social determinants (truly mulitlevel) are welcome.

Author Response

Reviewer 1:

This paper reports on a review of the National Cancer Institute's evidence-based cancer control programs (EBCCP) for breast cancer and provides a systematic approach to the evaluation of existing research captured by the EBCCP and gaps for identified risk factors.  The intent is to focus on the quality of existing research using defined criteria and on directing attention to the gaps.

General Comments

  1. The paper is well written and provides a useful review for breast cancer investigators interested in understanding where further research is needed to improve the quality of evidence or to fill gaps where they exist.

We thank the reviewer for these comments.

  1. Some attention to definitions and terminology may be helpful: a. 'Program' The authors are constrained by the terminology used by then NCI in naming the EBCCP, but are these really programs as might be expected in a program project, SPORE or center, rather than traditional research projects? Clarification of the meaning of 'program' in the EBCCPs is needed, b. 'Multilevel'. This term is used throughout the paper to describe interventions at the 'individual, interpersonal, community, and society' but it is not clear how the authors actually assessed program or project activities across all these levels. Clarification would help c. 'Health Disparities Populations'. I realize that this is the term used in the legislation (ref 14) and serves as a sort of shorthand,  but it is an awkward ungrammatical term that is meant to describe populations in which differences in health (cancer) outcomes or determinants have been observed and where health inequities are of concern. The author's term 'populations that face breast cancer disparities' or inequities is more satisfying as long as it is not used repeatedly.

We have clarified the criteria used by the NCI for selecting programs to be included in their EBCCP repository in the first paragraph of the Introduction (page 2):

“To be included in the repository, each EBCCP must have been published in a peer-reviewed journal; produced one or more positive behavioral and/or psychosocial outcome in a research study using an experimental or quasi-experimental design; and, unless otherwise copyrighted by the program developer, make available the program materials that include English for the larger cancer control community.”

We have also clarified how we defined and evaluated multilevel in the last paragraph of the Methods section (page 3):

“We assessed how many of the high-quality EBCCPs were multilevel (i.e., designed to address two or more levels of influence including the individual, household/family, school, clinic, and community levels) and were originally developed and implemented in populations facing breast cancer disparities. This includes diversity across race and ethnicity (defined as study population ≤50% NHW participants), socioeconomically disadvantaged participants, and/or those from medically underserved or rural geographic areas. We determined if the EBCCP was multilevel and developed in a population facing breast cancer disparities by reviewing the primary publications and, in some cases, the secondary publications listed on the NCI EBCCP website.”

Further, we agree with the reviewer about the prior wording and have replaced the term “health disparities populations” with “populations facing breast cancer disparities” throughout the manuscript.

  1. I question the strategic wisdom of suggesting that every risk factor needs an EBCCP (e.g., an EBCCP on breast feeding). This is a reductionist  and 'siloed' approach to a disease that is actually rather complex etiologically with multiple interactions over the lifecourse. Can the authors justify their or the NCI's motivations for such an approach?

We agree that breast cancer etiology is complex and multifactorial, and thus requires interventions that can address multiple risk factors simultaneously. We have added this important point to the Conclusions paragraph (page 9):

“Given that the etiology of breast cancer is complex and multifactorial, future programs are needed that move beyond the conventional approach of targeting individual risk factors in isolation and instead focus on multiple risk factors simultaneously.”

Specific Comments:

4.There is repetition between the last paragraph of the Introduction and paragraph 2.1 in the Methods. Would be best to put methods in the Intro into Methods.

We have condensed the last paragraph of the Introduction to avoid repetition with the Methods section. The specific details of our review process are now confined to the Methods section.

  1. In the footnote to Table 1 it refers to 'Health disparities population' as defined by greater than or equal to 50% non-Hispanic White. Shouldn't that be less than 50% White? Check that.

Yes. We have fixed this typo.

  1. Some of the language is overly dramatic. section 4.1 'grave concern', 'critical' this and that.

We have toned down this type of language throughout the manuscript.

  1. page 8, sect 4.2. The authors mention that there is no or little focus on cancer survival by EBCCPs, but everything preceding in Methods refers only to etiologic risk factors. If this is the intended focus of the EBCCPs it seems inappropriate to call it out as a gap.

We were referring to the fact that obesity and physical activity are also shown to reduce the risk of breast cancer recurrence and new primaries in breast cancer survivors, and thus these types of lifestyle interventions are also important for reducing further breast cancer risk in women with a personal history of breast cancer. We have clarified this point in the last paragraph of the Discussion section (page 9):

“For example, although some established risk factors, such as PA and maintaining a healthy body weight, are shown to improve outcomes after diagnosis [46,79-81], including breast cancer recurrence and new primaries [81,82], most programs were designed to reduce the risk of the first cancer and do not target cancer survivors. Instead, most interventions for cancer survivors largely focus on psychosocial interventions, which are important but might be complemented by lifestyle interventions.”

  1. Not a criticism or need for comment, but the discussion of the challenges faced by the assessment of body habitus and the weakness of the BMI as well as mention of the evolving challenges of understanding the influence of the microbiome and the need to move for research beyond the individual level to social determinants (truly mulitlevel) are welcome.

We appreciate this comment.

Reviewer 2:

Kehm and colleagues conducted a review of existing evidence-based cancer control programs (EBCCP) in the NCI’s online repository and identified research gaps and needs.  My comments are as follows:

My main comment relates to existing EBCCP’s on obesity.  13 EBCCPs on obesity were focused on young adults (19-39 years) and adults (≥40 years).  How many of the 13 were on young adults (19-39 years) and how many were for older adults (≥40 years)?   For the EBCCPs targeting those ages 19-39 years, I would have liked to know the specific messages on body weight/body mass index during young adult life in relation to risk of premenopausal breast cancer.  Five (17%) obesity EBCCPS met the quality criteria- and according to the discussion on page 8, 4 of these 5 obesity EBCCPS targeted childhood and adolescence.  I would have liked to know more about these 4 EBCCPS and their message since the authors stated that they  ‘may be effective for reducing premenopausal breast cancer”.   Perhaps a Table 3 should be added to further explain these 5 high-quality EBCCPs.    

We have added a breakdown of the number of studies targeting young adults (19-39 years), adults (40-65 years), and older adults (65+ years) in the Obesity subsection of the Results section (page 4):

“The remainder (n=13) focused on young adults (ages 19-39 years), adults (40-65 years), and/or older adults (65+ years) within work-based or other settings (e.g., religious institutions). Of these programs, one focused only on young adults, one focused only on adults, three focused only on older adults, three focused on young adults and adults, and five focused on young adults, adults, and older adults.”

We note that besides the one program that was developed for breast cancer survivors, which included all women older than 18 years of age, none of the programs were targeted towards women at increased risk for breast cancer. As far as we were able to tell based on our review, the programs targeting the general population did not include specific messaging about the link between obesity and premenopausal or postmenopausal breast cancer risk. We have added this point to the Obesity subsection of the Results section (page 4):

“Besides the one program targeting breast cancer survivors [18], none focused on populations at increased risk for breast cancer or, to the best of our knowledge, included specific messaging about the impact of obesity on breast cancer risk.”

We have removed the sentence in the Discussion section stating that obesity interventions in childhood and early adolescence are particularly important for reducing premenopausal breast cancer risk, and instead focus on the need for interventions in postmenopausal women. We also now discuss the importance of early life interventions for reducing the risk of obesity in adulthood, given that obesity tracks from childhood to adulthood. However, we also now make the point that the full impact of early life interventions on breast cancer risk reduction might be complicated by the fact that there is evidence of an inverse association between body fatness in early adulthood and breast cancer risk. Discussion section, pages 8-9:

“The lack of high-quality obesity programs in adult populations is a major gap given that postmenopausal women might benefit most in terms of breast cancer risk reduction from such programs, and multilevel interventions might prove more effective for groups with a greater prevalence of excess adiposity [74-76]. Youth programs targeting obesity are also important given that obesity tracks from childhood to adulthood [77,78]. However, understanding the full impact of early life obesity programs on breast cancer risk reduction is complicated by the fact that body fatness in early adulthood has been associated with decreased risk [8].”

I think the authors should consider having 2 separate entries under obesity or weight gain in Table 1.   The sentence “strong evidence that body size in young adulthood decreases pre and postmenopausal breast cancer”  is included as the second half of the sentence.  Although the author included a paragraph in their discussion on obesity, I believe more careful attention is needed in the messaging of the effect of ‘high body size in young adulthood’ on risk of pre and postmenopausal breast cancer.   

We now separate out “body fatness in young adulthood” and “body fatness and weight gain in adulthood” into two separate rows in Table 2.

Although I understand the appeal of suggesting interventions targeting adiposity and body composition measures, do we have the evidence to suggest recommendations in terms of WHR, percent body fat, fat mass index?  Will the recommendations be specific by age and race and ethnic groups?  

There is evidence from the World Cancer Research Foundation (WCRF) and American Institute for Cancer Research “Diet, Nutrition, Physical Activity and Cancer: A Global Perspective. Continuous Updated Project Expert Report 2018” (available online: https://www.wcrf.org/wp-content/uploads/2021/02/Summary-of-Third-Expert-Report-2018.pdf)that BMI, waist circumference, and waist-to-hip ratio in adulthood are all associated with increased risk of postmenopausal breast cancer. We have added this point to Table 2. We acknowledge that making recommendations about obesity interventions for breast cancer risk reduction is not necessarily straightforward, given that different associations have been found for obesity in early life versus adulthood. To get this point across, we have expanded our discussion of the relative importance of obesity programs in early life versus adulthood for breast cancer risk reduction: (pages 8-9):

“There was one high-quality program for youth that was multilevel and included a sample with >50% of individuals from populations facing breast cancer disparities [20], but there were none for adults. The lack of high-quality obesity programs in adult populations is a major gap given that postmenopausal women might benefit most in terms of breast cancer risk reduction from such programs, and multilevel interventions might prove more effective for groups with a greater prevalence of excess adiposity [74-76]. Youth programs targeting obesity are also important given that obesity tracks from childhood to adulthood [77,78]. However, understanding the full impact of early life obesity programs on breast cancer risk reduction is complicated by the fact that body fatness in early adulthood has been associated with decreased risk [8].”

Supplementary Table S1-  what are the 16 criteria under research integrity?

We have added the 16 criteria scored for research integrity to Supplementary Table S1.

Reviewer 2 Report

Kehm and colleagues conducted a review of existing evidence-based cancer control programs (EBCCP) in the NCI’s online repository and identified research gaps and needs.  My comments are as follows:

My main comment relates to existing EBCCP’s on obesity.  13 EBCCPs on obesity were focused on young adults (19-39 years) and adults (40 years).  How many of the 13 were on young adults (19-39 years) and how many were for older adults (40 years)?   For the EBCCPs targeting those ages 19-39 years, I would have liked to know the specific messages on body weight/body mass index during young adult life in relation to risk of premenopausal breast cancer.  Five (17%) obesity EBCCPS met the quality criteria- and according to the discussion on page 8, 4 of these 5 obesity EBCCPS targeted childhood and adolescence.  I would have liked to know more about these 4 EBCCPS and their message since the authors stated that they  ‘may be effective for reducing premenopausal breast cancer”.   Perhaps a Table 3 should be added to further explain these 5 high-quality EBCCPs.    

I think the authors should consider having 2 separate entries under obesity or weight gain in Table 1.   The sentence “strong evidence that body size in young adulthood decreases pre and postmenopausal breast cancer”  is included as the second half of the sentence.  Although the author included a paragraph in their discussion on obesity, I believe more careful attention is needed in the messaging of the effect of ‘high body size in young adulthood’ on risk of pre and postmenopausal breast cancer.   

Although I understand the appeal of suggesting interventions targeting adiposity and body composition measures, do we have the evidence to suggest recommendations in terms of WHR, percent body fat, fat mass index?  Will the recommendations be specific by age and race and ethnic groups?  

Supplementary Table S1-  what are the 16 criteria under research integrity?

Author Response

(The authors gave the same response as above.)
